# Antimicrobial resistance and molecular epidemiology of *Staphylococcus aureus* from Ulaanbaatar, Mongolia

Rajeshwari Nair[1], Blake M. Hanson[1], Karly Kondratowicz[1], Altantsetseg Dorjpurev[2], Bulgan Davaadash[2], Battumur Enkhtuya[2], Odgerel Tundev[2] and Tara C. Smith[3]

[1] Department of Epidemiology, College of Public Health, The University of Iowa, Iowa City, IA, United States
[2] Bacteriological Reference Laboratory, National Center for Communicable Diseases, Ulaanbaatar, Mongolia
[3] College of Public Health, The University of Iowa, Iowa City, IA, United States

## ABSTRACT

This study aimed to characterize *Staphylococcus aureus* (*S. aureus*) strains isolated from human infections in Mongolia. Infection samples were collected at two time periods (2007–08 and 2011) by the National Center for Communicable Diseases (NCCD) in Ulaanbaatar, Mongolia. *S. aureus* isolates were characterized using polymerase chain reaction (PCR) for *mecA*, PVL, and *sasX* genes and tested for *agr* functionality. All isolates were also *spa* typed. A subset of isolates selected by frequency of *spa* types was subjected to antimicrobial susceptibility testing and multilocus sequence typing. Among 251 *S. aureus* isolates, genotyping demonstrated methicillin resistance in 8.8% of isolates (22/251). Approximately 28% of the tested *S. aureus* isolates were observed to be multidrug resistant (MDR). Sequence type (ST) 154 (*spa* t667) was observed to be a strain with high virulence potential, as all isolates for this *spa* type were positive for PVL, had a functional *agr* system and 78% were MDR. *S. aureus* isolates of ST239 (*spa* t037) were observed to cause infections and roughly 60% had functional *agr* system with a greater proportion being MDR. Additionally, new multilocus sequence types and new *spa* types were identified, warranting continued surveillance for *S. aureus* in this region.

## INTRODUCTION

In the past fifty years, *Staphylococcus aureus* (*S. aureus*) has established itself as one of the most frequent antibiotic resistant bacterial pathogens in hospitals and communities (*Boucher & Corey, 2008*). *S. aureus* typically causes skin and soft tissue infections, but can also cause invasive infections such as bacteremia, sepsis, endocarditis, pneumonia, osteomyelitis, etc. (*Hidron et al., 2008*; *Liu et al., 2011*). In earlier years, *S. aureus* infections were commonly observed in individuals with a history of exposure to hospitals (*David & Daum, 2010*). There has been a major epidemiologic transition since the mid-1990s

Corresponding author
Tara C. Smith, tsmit176@kent.edu

when *S. aureus* was observed to cause infections in population with no known risk exposures. The emergence of such community-associated *S. aureus* has further magnified the challenge of *S. aureus* prevention and treatment practices (*David & Daum, 2010*). Methicillin-resistant *S. aureus* (MRSA) is associated with the rise in attributable mortality due to staphylococcal infections (*Cooper et al., 2004*).

Surveillance studies have observed a considerable difference in proportions of MRSA invasive infections in Europe ranging from <1% in Denmark and the Netherlands to 44% in the United Kingdom and Greece (*Cooper et al., 2004*; *Köck et al., 2010*). Nationwide surveillance for invasive MRSA infections conducted in the United States reported about 94,000 cases resulting in approximately 18,000 deaths (*Klevens et al., 2007*). Worldwide, rates of MRSA have been increasing as observed from data obtained via surveillance initiatives by the National Nosocomial Surveillance System (NNIS) and the European Antimicrobial Resistance Surveillance System (EARSS) (*Fridkin et al., 2002*; *Grundmann et al., 2006*; *Tiemersma et al., 2004*; *Turnidge & Bell, 2000*). Nevertheless, a major concern is the lack of data in many countries, particularly the developing countries, as this could potentially result in global transmission of undetected MRSA strains (*Azeez-Akande, 2010*; *Molton et al., 2013*).

Mongolia is a relatively small country in North-East Asia locked for the most part between China in the south and Russia in the north with a population of approximately 2.8 million (*Bataar et al., 2010*; *Ider et al., 2010*). The capital city of Ulaanbaatar is home to roughly half of the country's population. Infectious diseases still figure in the top 10 causes of death in the country, with sepsis being a common diagnosis among ICU patients (*Bataar et al., 2010*). The evolving political and economic changes in Mongolia have impacted the working of laboratory networks, data collection and management systems, and training of healthcare professionals in identification and prevention of hospital-acquired infections (*Ider et al., 2010*). There has also been a disruption of funds to hospitals to conduct surveillance for multidrug-resistant organisms. Due to insufficient laboratory capacity, hospitals in Mongolia use culture testing methods only when empiric therapy fails (*Ider et al., 2010*). This process could potentially propagate antimicrobial resistance in pathogens such as *S. aureus*. In addition, a large herder population in Mongolia with a livestock population of 43 million animals increases the risk of transmission of zoonotic infections (*WHO-Mongolia , 2010*).

A community-based survey conducted by the World Health Organization (WHO) in Ulaanbaatar between March and April 2009 observed a prevalence of 42% in the use of non-prescription antibiotics among children less than 5 years of age. This proportion is much higher than other regions such as rural communities in Vietnam (12%) and a Chinese city (36%) (*Togoobaatar et al., 2010*). The study found approximately 50% of the children in participating households were prescribed antibiotics, of which roughly 51% children were given both prescribed and non-prescribed antibiotics by their caregiver (*Togoobaatar et al., 2010*). In a developing country such as Mongolia, unconditional use of antibiotics is of particular concern (*Stefani & Goglio, 2010*). Selective pressures such as the unrestricted use of antibiotics and inadequate compliance to antibiotic regime in

conjunction with inadequate surveillance for antimicrobial resistance are some of the important reasons for the emergence of highly resistant *S. aureus* strains (*Grundmann et al., 2006*; *Stefani & Goglio, 2010*).

High population density, urbanization, inadequate infection control policies, exploding antibiotic use, and lack of appropriate healthcare delivery are some of the established social risk factors for colonization and transmission of *S. aureus* strains in hospitals and communities (*Charlebois et al., 2002*; *Chen et al., 2011*; *Clements et al., 2008*; *Henderson, 2006*; *Rehm & Tice, 2010*). There are very few studies in the published literature on the epidemiology of *S. aureus* in Mongolia. A study conducted in 2006 in Ulaanbaatar analyzed *S. aureus* infection isolates obtained from four university hospitals (*Orth et al., 2006*). Analysis using molecular methods and antibiotic susceptibility testing in isolates from this study determined the prevalence of MRSA to be very low (2.9%) (*Orth et al., 2006*). However, this study only included isolates collected between 2000 and 2002 and characterized only the six MRSA isolates identified in their cohort of *S. aureus* isolates. The aim of our study is to bridge the gap in *S. aureus* literature from Mongolia, and determine the *S. aureus* molecular epidemiology and antimicrobial resistance patterns in Mongolia.

## MATERIALS AND METHODS

This is an observational study conducted during two time periods (2007–08 and 2011) to investigate the prevalence of MRSA infections, and to characterize the *S. aureus* strains causing these infections in Mongolia. The University of Iowa IRB evaluated this project and determined that it did not qualify as human subjects research. To accomplish the study objective, we collaborated with the National Center for Communicable Diseases (NCCD) in Ulaanbaatar. The NCCD has an established Hospital Related Infection Surveillance and Research Unit (HRISRU) (*Ider et al., 2010*). This study characterized *S. aureus* isolated from human infections in a convenience sample obtained from the NCCD.

In 2007–08, the NCCD transported a convenience sample of "potentially confirmed" *S. aureus* isolates to the Centers for Emerging Infectious Diseases (CEID) at Coralville, IA as a collaborative project to characterize these infection isolates. Additionally, a study team member who traveled to Mongolia in 2011 tested potential *S. aureus* infection isolates banked in Ulaanbaatar and shipped back frozen isolates to the CEID. Patient information such as age, gender, and sample type were available only for the 2007–08 infection samples. Isolates were labeled "wound" if collected from surgical site infections or wound samples. All urine samples collected in 2007–08 were obtained from voided urine, i.e., none of these patients were catheterized.

### Biochemical testing and DNA isolation

Isolates were grown and re-confirmed to be *S. aureus* at the CEID, as described previously (*O'Brien et al., 2012*). All *S. aureus* isolates were frozen and stored in glycerol broth solution at −80°C for future use. *S. aureus* DNA was extracted using the Wizard Genomic DNA Purification Kit (Promega, Madison WI) following manufacturer's instructions.

## Antimicrobial susceptibility testing (AST)

The antimicrobial susceptibility of isolates were tested by the broth microdilution method in accordance with the Clinical Laboratory Standards Institute (CLSI) standards (*CLSI, 2012*). Isolates were tested for susceptibility to the following 11 antimicrobials: oxacillin, gentamicin, erythromycin, clindamycin, tetracycline, trimethoprim/sulfamethaxozole (TMP/SMX), imipenam, levofloxacin, linezolid, vancomycin, and daptomycin. Resistance to high-level mupirocin and inducible clindamycin resistance (ICR) were also examined. All AST-confirmed MRSA isolates were considered to be multidrug resistant (MDR). MSSA isolates non-susceptible to ≥1 antimicrobial agent in ≥3 discrete antimicrobial categories were also classified as MDR, as per a recently published report on standardization of bacterial antimicrobial resistance profiles (*Magiorakos et al., 2012*).

### *S. aureus* genetic analysis

Amplification of the *spa* fragment was performed using methods and primers described previously (*Shopsin et al., 1999*). Identification of the *spa* type for each isolate, the Based Upon Repeat Pattern (BURP) analysis to identify *spa* cluster complexes (*spa*CCs), and calculation of the diversity index (*Hunter & Gaston, 1988*) and corresponding confidence intervals (*Grundmann, Hori & Tanner, 2001*) was performed using the Ridom StaphType software (version 2.2.1; Ridom GmbH, Würzburg, Germany) (*Harmsen et al., 2003*; *Mellmann et al., 2007*; *Mellmann et al., 2008*; *Strommenger et al., 2008*). All isolates were tested for the Panton-Valentine leukocidin (PVL) (*luk*S-PV and *luk*F-PV) (*Lina et al., 1999*), *mecA* (*Boşgelmez-Tınaz et al., 2006*), and the *sasX* gene as previously described (*Holden et al., 2010*; *Li et al., 2012*). Identified positive and negative controls were used in all molecular assays.

### Accessory gene regulator (*agr*) testing

*agr* functionality (functional or dysfunctional) was measured using the level of δ-hemolysin production, as described previously (*Sakoulas et al., 2002*; *Schweizer et al., 2011*; *Traber & Novick, 2006*). *agr* positive and negative reference strains were used as controls to ensure validity of our findings.

### Multi-locus sequence typing (MLST)

The allelic profile of *S. aureus* isolates were determined as described previously (*Enright et al., 2000*). At least one representative isolate from the most frequent *spa*CCs, rare *spa*CCs, *spa* types such as t037 that could potentially have two sequence types, or when available a MRSA isolate was selected for MLST analysis from each time period.

## STATISTICAL ANALYSIS

Data analysis was performed using the SAS statistical software (Version 9.3, SAS Institute Inc., Cary, NC). We used the 2-tailed Fisher's exact test, and the Wilcoxon signed-rank test to analyze categorical and continuous variables, respectively. *P* values ≤0.05 were considered statistically significant for associations between explanatory variables such as age, gender, and type of infection and *S. aureus spa* type, *mecA*, PVL

**Table 1  Prevalence of *S. aureus* genes in Mongolia MRSA and MSSA isolates.**

| Factor tested | 2007–08 (*N* = 53) | | | 2011 (*N* = 198) | | |
|---|---|---|---|---|---|---|
| | **MSSA** | **MRSA** | ***p*-value** | **MSSA** | **MRSA** | ***p*-value** |
| **PVL** | | | | | | |
| Positive | 40 (85.1) | 5 (83.3) | 1.00 | 55 (30.2) | 8 (50) | 0.16 |
| Negative | 7 (14.9) | 1 (16.7) | | 127 (69.8) | 8 (50) | |
| ***agr*[a]** | | | | | | |
| Functional | 37 (86.1) | 2 (66.7) | 0.39 | 175 (96.2) | 13 (81.3) | 0.037 |
| Dysfunctional | 6 (13.9) | 1 (33.3) | | 7 (3.9) | 3 (18.8) | |

**Notes.**

[a] Seven *S. aureus* isolates from 2007–08 did not grow for *agr* testing.
Significant if $p \leq 0.05$.
Data for $2 \times 2$ table presented as frequency (%).
MRSA, methicillin-resistant *S. aureus*; MSSA, methicillin-susceptible *S. aureus*.

and *agr* functionality. Antimicrobial susceptibility results were analyzed by year of data collection and methicillin-resistance status to observe trends in resistance for each tested antibiotic. Association between *S. aureus* antimicrobial susceptibility, MDR, *mecA*, PVL, *agr* functionality, and *spa* types were also assessed. Odds ratio (OR) and 95% confidence interval (CI) was reported for significant associations.

# RESULTS

## Patient and sample characteristics

In total we analyzed 252 potential *S. aureus* isolates, 198 from 2011 and 54 from 2007–08 isolate collections. Of these, 251 were confirmed to be *S. aureus* isolates. Patient demographics and sample characteristics were available only for the 53 isolates collected in 2007–08. The age of patients ranged from 1 day–82 years (median: 24 years). Of the 53 patients, 31 (58.5%) were females and 22 (41.5%) males. Approximately 43% of the *S. aureus* were isolated from wound samples.

## Molecular typing

We observed a greater proportion of isolates from the 2007–08 collection to be positive for the *mecA* gene. The cohort of *S. aureus* isolates from 2007–08 had greater proportion of PVL positivity in both the MSSA and MRSA isolates, with 59% of MRSA isolates being positive for the PVL gene (Table 1). *S. aureus* isolates were observed to have a greater proportion of functional *agr* in both time periods, albeit the proportions were higher for MSSA and MRSA in the 2011 collection. We observed that *mecA* positive *S. aureus* isolates had a 77% less likelihood of having a functional *agr* (OR = 0.23, 95% CI 0.067, 0.79; $p = 0.033$). A borderline significance was observed between age and PVL positivity in that most of our PVL-positive *S. aureus* isolates were obtained from older individuals ($p = 0.045$, other data not shown). All isolates tested negative for the *sasX* gene.

## Antimicrobial susceptibility patterns

A subset of *S. aureus* isolates (80/251, ~32%) were tested for antimicrobial susceptibility, based on the frequency of *spa* types. The proportion of *S. aureus* isolates that were MRSA (MDR) was greater in 2011 (Fig. 1A). Interestingly, we did not observe any *S. aureus* isolates that belonged to the MSSA-MDR category in 2011. Isolates in both years have comparable proportions of MSSA that do not meet the MDR criteria. There was no significant difference in the proportion of MDR isolates between the two study periods ($p = 0.092$).

Overall, the proportion of antibiotic resistance in tested *S. aureus* isolates was 38.8%. Greater proportion of MRSA isolates in 2011 was observed to be resistant to erythromycin, tetracycline, levofloxacin and TMP/SMX (Fig. 1B). We also observed a wider spectrum of resistance in the 2011 MRSA isolates as there is additional resistance to gentamicin, imipenam, and ICR. MSSA isolates were observed to be proportionately more resistant to erythromycin and tetracycline in 2011 while resistance to levofloxacin, erythromycin and TMP-SMX was observed in 2007 isolates (Fig. 1C). We did not identify high-level mupirocin resistance in our isolates. There was good concordance in *mecA* positivity and phenotypic expression of oxacillin resistance among tested *S. aureus* isolates. Three isolates were observed to have discordant oxacillin-resistance phenotype-genotype (data not shown).

We observed that *S. aureus* isolates resistant to 3 or more antimicrobials had significantly lower odds of having a functional *agr* system (OR = 0.166, 95% CI 0.046-0.59, $p = 0.014$). Of the 22 MDR isolates, roughly 55% (12/22) were positive for the PVL gene, about 86% (19/22) positive for the *mecA* gene, and approximately 64% (14/22) had a functional *agr* phenotype.

## *spa* type distribution (BURP) and multilocus sequence type (MLST)

Eleven confirmed *S. aureus* isolates were identified as "non-typeable" after at least two attempts to sequence the *spa* gene. Of these, 8 isolates demonstrated bands on amplification of the *spa* fragment with the published forward primer as well as an alternate forward primer (*Molla et al., 2012*). However, the RIDOM StaphType software could not assign *spa* types, possibly due to deviating repeats as demonstrated in previous studies (*Baum et al., 2009*; *Sörum et al., 2013*). The remaining 3 isolates demonstrated double bands for the *spa* gene as observed in a study conducted by *Shakeri et al. (2010)*, which could not give *spa* types even after three attempts on PCR or purification of bands from an agarose gel.

We identified 63 distinct *spa* types among the 251 *S. aureus* isolates (Table 2). The most common *spa* types in 2007–08 were t589 (13%), t3465 (13%), and t435 (11%). *spa* types t435 (10%), t589 (8%), t5288 (7.5%), and 7% each t1460 and t8677 were the most frequently occurring strains in 2011. *spa* types belonging to *spa*CC 667 was observed to be the most frequently isolated MRSA strain while MSSA isolates were frequently of the *spa*CC 589 type. *S. aureus* isolates of *spa* types t589, t435, t037 and t667 were observed

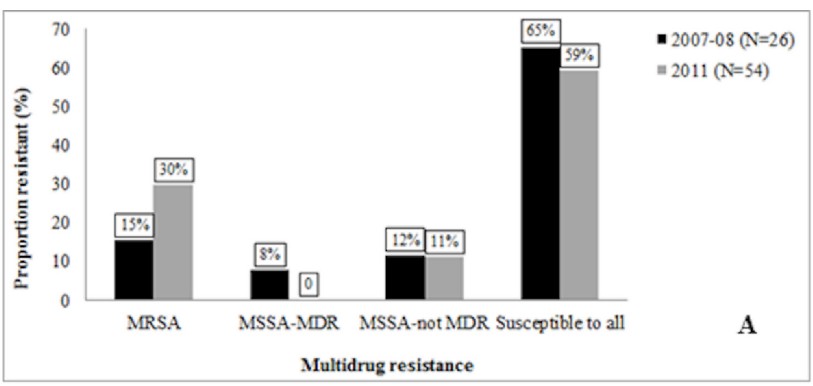

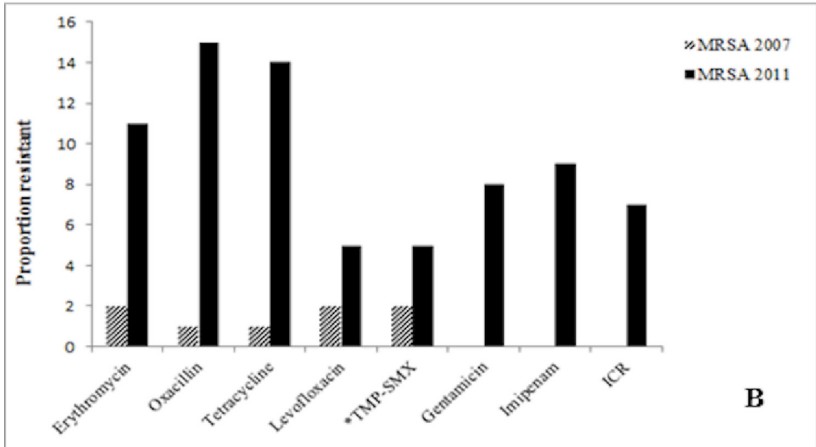

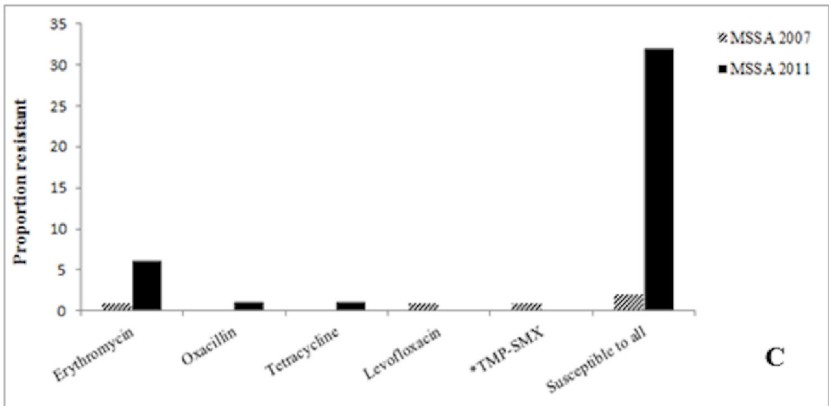

**Figure 1 Antimicrobial resistance in Mongolia *S. aureus* isolates.** (A) Illustration for multidrug resistance in Mongolia *S. aureus* isolates. MRSA, methicillin-resistant *S. aureus*; MSSA, methicillin-susceptible *S. aureus*; MDR, multidrug resistant; MSSA-not MDR category include MSSA isolates that are non-susceptible to ≥1 antimicrobial agent in (continued on next page...)

**Figure 1 (...continued)**

<3 discrete antimicrobial categories. (B) Graph for antimicrobial resistance in MRSA isolates. *TMP-SMX, trimethoprim/sulfamethaxozole; ICR, inducible clindamycin resistance. Resistance to rifampicin and complete resistance to clindamycin were not observed. (C) Graph for antimicrobial resistance in MSSA isolates. No resistance was observed to gentamicin, rifampicin, imipenam, clindamycin or inducible clindamycin resistance.

**Table 2** Distribution of *spa* types and *spaCC* among Mongolia MRSA and MSSA isolates.

**(A) MRSA**

| *spaCC* | Study assigned *spaCC* | No. (%) of strains | *spa* types[a] |
|---|---|---|---|
| *spa*CC 589 | CC1 | 2 (9.1) | **t589**, t2397 |
| *spa*CC 435 | CC2 | 1 (4.5) | t435 |
| *spa*CC 037 | CC4 | 5 (22.7) | **t037**, t074 |
| *spa*CC 667 | CC7 | 8 (36.4) | **t667**, t2832 |
| *spa*CC 8 | CC9 | 1 (4.5) | **t008** |
| **Total** | | **17** | |

**(B) MSSA**

| *spaCC* | Study assigned *spaCC* | No. (%) of strains | *spa* types[b] |
|---|---|---|---|
| *spa*CC 589 | CC1 | 76 (33.2) | **t5288**, t3126, t1460, t589, t4153, t6242, t3103, t630, t073, t3219, t7043, t102, t550, t722, t908, t10064[c] |
| *spa*CC 435 | CC2 | 64 (27.9) | **t435**, t270, t2392, t169, t272, t284, t159, t308, t8762, t6870, t2087, t1441 |
| *spa*CC 8677 | CC3 | 34 (14.8) | **t8677**, t4049, t3156, t10066[c], t4473, t3465 |
| *spa*CC 037 | CC4 | 5 (2.2) | **t037**, t021 |
| *spa*CC 084 | CC5 | 4 (1.7) | **t084**, t085, t1038 |
| *spa*CC 1194 | CC6 | 3 (1.3) | **t1194**, t1710 |
| *spa*CC 667 | CC7 | 3 (1.3) | t667 |
| *spa*CC 1010 | CC8 | 2 (0.9) | **t1010**, t081 |
| *spa*CC 8 | CC9 | 1 (0.4) | t024 |
| **Total** | | **192** | |

**Notes.**

[a] Bolded *spa* type is the putative founder for that *spaCC*.
BURP clusters formed of 22 MRSA isolates.
3 non-typeable isolates and one *spa* type (2 isolates) excluded from BURP clusters.
MRSA, methicillin-resistant *S. aureus*.

[b] Bolded *spa* type is the putative founder for that *spaCC*.

[c] Identified as new *spa* types.
BURP clusters formed of 229 MSSA isolates.
8 non-typeable isolates and four *spa* types (6 isolates) excluded from BURP clusters.
MSSA, methicillin-susceptible *S. aureus*.

**Table 3 Diversity index for Mongolia MRSA and MSSA isolates by *spa* typing and BURP.**

| Organism groups | No. of isolates | No. of isolates rejected by BURP | Number of different spa types | Typeability (%) | Diversity/discriminatory index (95% CI) |
|---|---|---|---|---|---|
| MRSA (n = 22) | 19 | 3 | 8 | 95 | 0.865 (0.767–0.964) |
| MSSA (n = 229) | 222 | 7 | 58 | 99.11 | 0.952 (0.942–0.962) |
| MRSA + MSSA (n = 251) | 241 | 10 | 63 | 98.77 | 0.956 (0.947–0.965) |

**Notes.**

CI, Confidence Interval; BURP, Based Upon Repeat Pattern analysis using the RIDOM StaphType; MRSA, methicillin-resistant *S. aureus*; MSSA, methicillin-susceptible *S. aureus*.

in both MRSA and MSSA groups (Tables 2A and 2B). We identified 8 different *spa* types among the MRSA isolates and 58 *spa* types among the MSSA isolates. The diversity index was higher for MSSA isolates (0.952) compared to MRSA isolates (0.865), but due to the small sample size of MRSA isolates the confidence intervals overlap (Table 3).

BURP analysis revealed clustering around putative founder *spa* types t589 (2011), t435 (2007–08 and 2011), t3465 (2007–08), and t8677 (2011) (Fig. 2). We identified three new MSSA *spa* types (t10064, t10066, and t10358) in the 2011 collection. *spa*CC 667 constituted only 5% of all the strains. Nevertheless, isolates in this group had high PVL prevalence (100%), high *mecA* prevalence (72.7%), 100% functional *agr* isolates, and high multi-drug resistance. *spa*CC 037 also had high multidrug resistance but had lower presence of PVL (10%), *mecA* (50%) and functional *agr* (60%). Singleton *spa* types were t002, t126, t156, t521, t647, t803, t1451, t3329 and t8039. None of the tested singletons appeared to be MDR.

A subset of *S. aureus* isolates from each time period was tested by MLST revealing the following information: t021 (ST30), t037 (ST239), t084 (ST15), t435 (ST121) and t667 (ST154). *spa* types t1460, t5288, and t589 were observed to be ST45 in our study. One t589 MRSA isolate from the 2011 collection was identified to be a new sequence type ST2600, a double-locus variant (DLV) of ST45 as indicated by MLST. In addition, isolates belonging to *spa* types t8677 and t3465 in our sample set have novel *arcC* and *aroE* alleles and was categorized as ST2737 by the MLST database curator.

## DISCUSSION

The prevalence of MRSA among clinical *S. aureus* isolates obtained from Mongolia was 8.8%. A study conducted in 2006 in Mongolia analyzing *S. aureus* isolates collected between 2000–02 found a low prevalence of MRSA (2.9%) by susceptibility testing (Orth et al., 2006). Our study is based on a convenience sample of Mongolian *S. aureus* isolates. Nevertheless, we could infer with caution that there may be an increase in the prevalence of MRSA in Mongolia, reflected by the increase in MDR-MRSA with the relative absence of MDR-MSSA isolates in our sample set. Due to lack of published *S. aureus* data from Mongolia we compared our results to studies from China and Russia since there is a potential for transmission given its geographical proximity. China reported a prevalence of

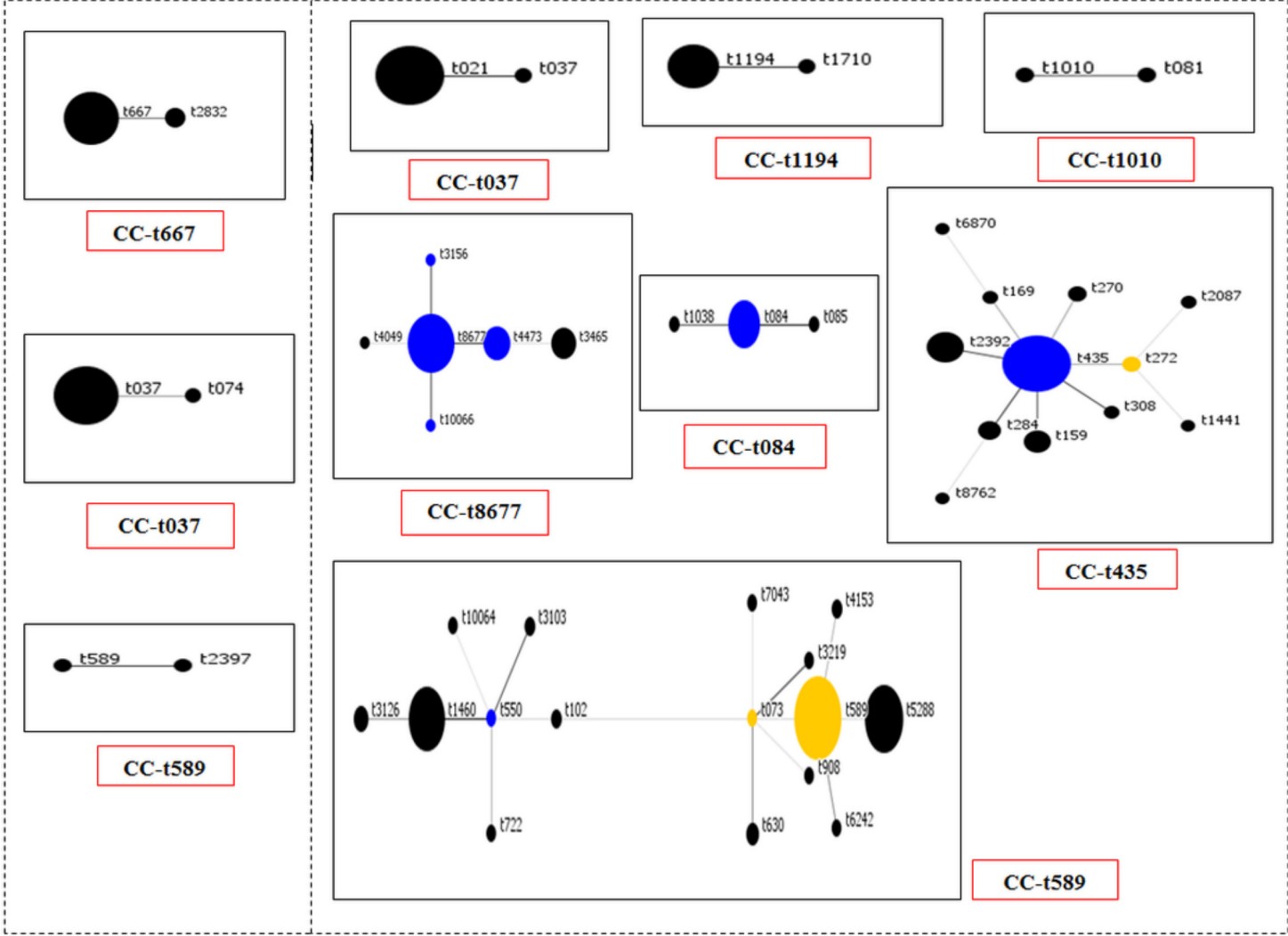

**Figure 2 Population snapshot for MRSA & MSSA Based-Upon Repeat Pattern (BURP) analysis.** BURP grouping using default parameters resulted in 8 *spa*CCs and excluded 5 *spa* types (t026, t132, t517, t2493, and t10358). Each dot represents a unique *spa* type. Diameter of a dot is proportional to the quantity of corresponding *spa* type. Blue dots, group putative founders (i.e., *spa* type with the highest score within the CC). Yellow dots, putative subfounders with second highest score. If two or more *spa* types have the same highest founder score they are illustrated in blue. The distance between linked and/or unlinked *spa* types do not concern the genetic distance between them.

50.4% for MRSA in 2005 with considerable variations even within the country (*Chu et al., 2013*; *Song et al., 2013*; *Wang et al., 2008*; *Yu et al., 2012*; *Zhao et al., 2012*). The proportion of methicillin resistance reported from *S. aureus* in Russia varied from 18% (*Vorobieva et al., 2008*) to 48% (*Baranovich et al., 2010*). Our study data observed a lower prevalence of MRSA relative to the neighboring countries. Nevertheless, given the convenience sample this may be an underestimate of the 'true' prevalence of MRSA in Mongolia.

Our study observed genetic diversity in the MSSA isolates compared to the MRSA isolates. However, we note that the observed diversity could not be concluded to be statistically significant given the small sample size of MRSA isolates resulting in overlap of confidence intervals. A limited number of *spa* types were common in both groups and these are

not genetically unrelated strains. This observation of common *spa* types is consistent with findings from other studies (*Hallin et al., 2007*; *Strommenger et al., 2008*), and supports the view that MRSA could potentially emerge from existing MSSA clones by acquisition of the SCC*mec* complex (*Enright et al., 2002*; *Robinson & Enright, 2003*). It is more likely, however, that in Mongolia, MRSA infections are increasing in prevalence due to strain replacement, not SCC*mec* acquisition, which is supported by the presence of MDR-MRSA without any MDR-MSSA in 2011. We are unable to conclude this hypothesis of strain replacement with certainty, hence it is crucial to implement surveillance protocols for *S. aureus* in Mongolia, as there are a myriad of factors contributing to antimicrobial resistance.

We observed a significant association between antimicrobial resistance and functionality of the *agr* system suggesting a potential influence of antimicrobial resistance on the fitness of the pathogen via the *agr* system, or vice versa (*Paulander et al., 2013*). In addition, there appears to be a significant association between presence of the gene for methicillin resistance and having a dysfunctional *agr* system. Evidence from our study on the impact of drug resistance on the regulation of *S. aureus* virulence is consistent with findings from previous studies that observed changes in *S. aureus* cell wall induced by methicillin. This process was observed to affect the bacterial quorum sensing system leading to reduced virulence by suppression of toxins (*Hao et al., 2012*; *Rudkin et al., 2012*). These findings could potentially influence treatment options for *S. aureus* infections in Mongolia by considering the trade-off between fitness of the strain and its range of antimicrobial resistance.

Our data suggests that the MRSA clone ST239-*spa* t037 is being transmitted amongst the population. This MLST type was also reported in the previous study from Mongolia, suggesting that ST239 could potentially be the dominant MRSA clone circulating in the country (*Orth et al., 2006*). ST239, a *S. aureus* bacterial hybrid formed by the admixture of MRSA clonal complexes ST30 and ST8 has been reported to be the dominant hospital clone in Asia (*Aires de Sousa et al., 2003*; *Baranovich et al., 2010*; *Song et al., 2013*; *Xu et al., 2009*; *Yamamoto et al., 2012*; *Yu et al., 2012*), Europe (*Alp et al., 2009*; *Szczepanik et al., 2007*; *Wisplinghoff et al., 2005*), South America (*Carvalho, Mamizuka & Gontijo Filho, 2010*; *Vivoni et al., 2006*), and the Middle East (*Cîrlan et al., 2005*) and even responsible for an outbreak of device-associated bacteremia in Europe (*Edgeworth et al., 2007*). In accordance with previous reports, all three identified ST239 *S. aureus* strains in our study were observed to be MDR (*Smyth et al., 2010*). We also observed *S. aureus* sequence types ST45 and ST121, which have previously been observed to be associated with *S. aureus* infections in other regions of the world, circulating in Mongolia.

The proportion of multidrug resistance was observed to be higher in 2011 relative to 2007–08, albeit this difference was not statistically significant. Reports on *S. aureus* multidrug resistance observed variable rates ranging from ~29%–100% from China (*Chao et al., 2013*; *Chen et al., 2009*; *Wang et al., 2012*) and about 90% in Russian MRSA isolates (*Baranovich et al., 2010*).

Studies observed that presence of the *sasX* gene in *S. aureus* potentially increased its virulence capacity by boosting the bacterial defense mechanism, particularly in the ST239

clones (*Li et al., 2012*). Furthermore, testing of *S. aureus* isolates from three teaching hospitals in eastern China demonstrated that the proportion of *S. aureus* strains that were non-ST239 and positive for the *sasX* gene increased from 5% to 28% between 2003–05 and 2009–11 (*Holden et al., 2010*; *Li et al., 2012*). None of our isolates, including the identified ST239 isolates, exhibited the presence of the *sasX* gene, suggesting its prevalence may be low or absent in Mongolia. We did not identify any known livestock-associated strains in our isolate collection, although several reports from China have observed the presence of these strains (*Wagenaar et al., 2009*; *Zhao et al., 2012*).

Our study has several limitations. *S. aureus* isolates were collected as a convenience sample and could not be consistently linked to important patient information, particularly the 2011 collection that had a larger sample size for *S. aureus* isolates. This could have potentially resulted in inclusion of duplicate patient isolates and over-representation of *S. aureus* strains. In addition, isolates from both time periods were not collected in a systematic manner adding to the potential selection bias. Hence, results from this study may reflect only a snapshot of the 'true' estimate of *S. aureus* infections in Mongolia. Conclusions drawn from this study could be used as preliminary results to design studies that are sufficiently powered to validate the observed associations. Nevertheless, there are not many studies from Mongolia and our study adds valuable information on the molecular epidemiology of *S. aureus* infections in Mongolia.

Another drawback of our study was the inability to differentiate the potential origin of *S. aureus* strains as healthcare associated (HCA-) versus community associated (CA-) since we did not have access to the date of admission before *S. aureus* isolation from the infection. Given that in recent times there has been a gradual blurring in the origin of *S. aureus* strains, the reliability of this differentiation may be questionable (*Mera et al., 2011*). In addition, we also did not test all *S. aureus* isolates in our collection for antimicrobial susceptibility that could have resulted in loss of information on 'true' antimicrobial susceptibility trends.

## CONCLUSIONS

In summary, our study observed an increasing prevalence of MRSA (8.8%) by AST, and recorded MDR rate of 28% in *S. aureus* isolates from Mongolia. We also observed the presence of previously identified *S. aureus* strains such as ST239 and ST30 adding its virulence potential to an existing burden of antimicrobial resistance. Regular surveillance and implementation of stricter policies for antimicrobial use is warranted to prevent further transmission of *S. aureus* in Mongolia.

## ACKNOWLEDGEMENTS

We thank Dr. Michael Otto and his lab for kindly providing the *sasX* positive control for molecular analysis, Dr. Marin L. Schweizer for sharing her expertise on the agr system and her time training a graduate student on agr testing methods, and Dr. Brett Forshey for his assistance and inputs with statistical analysis. We also acknowledge and thank our collaborators at the NCCD in Mongolia for their assistance with sample collection, sample shipping, assimilation and communication of patient data, and review of the manuscript.

### Funding

This study was funded by start-up funds from the University of Iowa (TCS). KK was supported in part by a Stanley Graduate Award for International Research. The funders had no role in study design, data collection and analysis, decision to publish, or preparation of the manuscript.

### Grant Disclosures

The following grant information was disclosed by the authors:
University of Iowa.
Stanley Graduate Award for International Research.

### Competing Interests

Tara C. Smith is an Academic Editor for PeerJ.

### Author Contributions

- Rajeshwari Nair and Blake M. Hanson performed the experiments, analyzed the data, wrote the paper.
- Karly Kondratowicz conceived and designed the experiments, performed the experiments, wrote the paper.
- Altantsetseg Dorjpurev, Bulgan Davaadash and Battumur Enkhtuya performed the experiments, provided clinical data.
- Odgerel Tundev conceived and designed the experiments, performed the experiments, provided clinical data.
- Tara C. Smith conceived and designed the experiments, wrote the paper.

### Ethics

The following information was supplied relating to ethical approvals (i.e., approving body and any reference numbers):

This study was submitted for a Human Subjects Research Determination (HSRD) by the University of Iowa IRB and was determined to be exempt from HSR and IRB approvals as no identifying information was collected.

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
