# Peer review of "Antimicrobial resistance and molecular epidemiology of Staphylococcus aureus from Ulaanbaatar, Mongolia"

_PeerJ, doi:10.7717/peerj.176_

## Round 0.1 · original submission · Major Revisions

Please follow all the comments, in particular of Reviewer 1, who suggested more imprtant changes than reviewer 2.

Reviewer 1 ·

Basic reporting

Some of the references used in the Introduction and Discussion should be revised because there are not the mos appropriate.

Experimental design

The authors should better describe how were the samples obtained and transported. where were the different methods used or applied.

Validity of the findings

Please check general comments

Additional comments

I think that it is essential that the authors separate MRSA from MSSA and then describe the characterization of each of the classes. Antibiotypes, ST and spa types, presence of PVL are meaningful if presented in the way I suggest. Both in the text and in the Tables and figures this separation is required. The Authors can then compare the data obtained with results from other countries namely China.

Concerning frequency of sarX among ST239 the authors need to revise the numbers. For example in the paper quoted from Holden et al 2010 I do not find any mention to the frequency of sarX.

Reviewer 2 ·

Basic reporting

No Comments

Experimental design

No Comments

Validity of the findings

See below for specifics.

Additional comments

This manuscript describes a characterization of Staphylococcus aureus from Mongolia. The study was well-designed and aware of limitations. The authors need to clarify some statements and results, and they need to reconsider some of their interpretations, as described below.

1) It is not clear whether the association between mecA(or methicillin or antimicrobials?) and agr function is positive or negative? The wording is poor. Egs, pg7, ln181; pg8,ln199; pg10, ln245-249. How does their finding compare with that described by Rudkin et al. 2012 J Infect Dis 205:798?

2) Pg8, ln204, says 11 "confirmed" non-typeable spa sequences. Does this mean they got no amplicon upon repeated PCR attempts, or they got a novel spa sequence? If the former, then this is an unexpectedly high number of isolates, and may call for a more thorough examination of the result (eg different PCR conditions, different primers).

3) Pg9, ln210 and Pg10, ln240, the authors note a "greater diversity" among the MSSAs. Three points here: 1) Are they simply equating number of spa types with diversity? They have not statistically compared diversity between MSSA and MRSA. Number of types can follow sample size, so it may be better to quantify diversity with an "index" (eg Simpson's) that incorporates number of types and frequency of occurrence, along with an appropriate confidence interval. 2) Many references exist that describe more "diversity" among MSSA; some should be cited. 3) Finding more diversity among MSSAs than MRSAs does not necessarily support their statement about MRSAs arising from exisiting MSSAs (ln241-242) - it depends on whether the MRSA types are a subset of the MSSA types. They could be completely different types yet of different "diversity".

4) Pg10, ln252-254, Because they find some of the same MLST-STs in the two time periods they suggest there may be minimal or absent mutations in S. aureus core genome? This logic is flawed. MLST examines only a small part of the genome, so there could be dozens to hundreds of SNPs outside of MLST genes that are not detected. In addition, spa t589 was noted to include both ST45 and "new ST" - is the "new ST" simply a single bp variant of ST45 or is it something quite different? If it is a new, single bp variant of ST45, then they have evidence against their statement of minimal/absent mutations.

5) It is also not clear what the strategy was for selecting isolates for MLST. Eg the spa t8677 group was the 3rd most common, yet no MLST was done; several rare spa types were done with MLST. One of the more interesting results is that ST45 and ST121 might be quite common in Mongolia; they are not so common elsewhere. Also could the spa types in Table 2 that include MRSAs be indicated with asterisks (or 2 asterisks if solely MRSA)?


Comments of a minor nature:

6) In the abstract, when you say "a high multidrug resistance profile" do you mean high level MICs to at least 3 antibiotics or resistance to many more than 3 antibiotics?

7) The quality of the Mongolia 2013 reference is not clear; suggest delete from the paper (pg3, ln72, as you already have 2 quality references for that statement).

8) On pg 5, ln135, when you say ">= 3 discrete antimicrobial categories" do you include oxacillin and imipenam in the same category, they are both beta-lactams but different subclasses. All the other antibiotics are different classes.

9) On pg6, ln150, it is probably worth indicating that the study reporting a high prevalence of sasX focused on isolates from (part of) China. We know very little about the geographic distribution of this gene.

10) Pg6, ln153, indicate how many replicates were done for the agr functionality assay per isolate.

11) Pg7, ln170, it is not clear that patients were "enrolled" as this study did not have human subjects involvement with consent forms, etc. Possibly the authors mean to say "The patients from which the isolates had been collected as part of routine microbiological work..." or something like that.

12) Pg9, ln209 and Table 2 and Fig 2 legends. It is better to call it a "putative" founder, as this BURP algorithm has not been rigorously examined for accuracy in identifying founders.

13) Pg11, ln277, does "unable to identify any duplicate isolates" mean they did not have the information to do so (eg which isolates are from which patients) or does it mean they did not identify any duplicates?

14) Pg12, ln292, 38.8% should be 8.8% as stated in the Abstract?

---

## Round 0.2 · accepted · Accept

The paper is now much more focused and it reads better. All the reviewer suggestions have been complied resulting in a great improvement on the structure of the manuscript. It should be accepted for publication.